# Elevated UTI Biomarkers in Symptomatic Patients with Urine Microbial Densities of 10,000 CFU/mL Indicate a Lower Threshold for Diagnosing UTIs

**DOI:** 10.3390/diagnostics13162688

**Published:** 2023-08-16

**Authors:** Laura K. S. Parnell, Natalie Luke, Mohit Mathur, Richard A. Festa, Emery Haley, Jimin Wang, Yan Jiang, Lori Anderson, David Baunoch

**Affiliations:** 1Department of Scientific Writing, Precision Consulting, 6522 Harbor Mist, Missouri City, TX 77459, USA; lksparnell@yahoo.com; 2Department of Clinical Research, Pathnostics, 15545 Sand Canyon Suite 100, Irvine, CA 92618, USA; nluke@pathnostics.com (N.L.); ehaley@pathnostics.com (E.H.); 3Department of Medical Affairs, Pathnostics, 15545 Sand Canyon Suite 100, Irvine, CA 92618, USA; mmathur@pathnostics.com; 4Department of Research and Development, Pathnostics, 15545 Sand Canyon Suite 100, Irvine, CA 92618, USA; rfesta@pathnostics.com; 5Department of Statistical Analysis, Stat4Ward, 2 Edgemoor Lane, Pittsburgh, PA 15238, USA; bob.wang@stat4ward.com (J.W.); ellie.jiang@stat4ward.com (Y.J.); 6Department of Diagnostic Market Access, Pathnostics, 15545 Sand Canyon Suite 100, Irvine, CA 92618, USA; landerson@pathnostics.com

**Keywords:** urinary tract infection (UTI), standard urine culture (SUC), multiplex polymerase chain reaction (M-PCR), urine biomarkers, diagnostic testing, neutrophil gelatinase-associated lipocalin (NGAL), interleukin (IL)-8, IL-1β

## Abstract

The literature lacks consensus on the minimum microbial density required for diagnosing urinary tract infections (UTIs). This study categorized the microbial densities of urine specimens from symptomatic UTI patients aged ≥ 60 years and correlated them with detected levels of the immune response biomarkers neutrophil gelatinase-associated lipocalin (NGAL), interleukin-8 (IL-8), and interleukin-1-beta (IL-1β). The objective was to identify the microbial densities associated with significant elevation of these biomarkers in order to determine an optimal threshold for diagnosing symptomatic UTIs. Biobanked midstream voided urine samples were analyzed for microbial identification and quantification using standard urine culture (SUC) and multiplex-polymerase chain reaction (M-PCR) testing, while NGAL, IL-8, and IL-1β levels were measured via enzyme-linked immunosorbent assay (ELISA). NGAL, IL-8, and IL-1β levels were all significantly elevated at microbial densities ≥ 10,000 cells/mL when measured via M-PCR (*p* < 0.0069) or equivalent colony-forming units (CFUs)/mL via SUC (*p* < 0.0104) compared to samples with no detectable microbes. With both PCR and SUC, a consensus of two or more elevated biomarkers correlated well with microbial densities > 10,000 cells/mL or CFU/mL, respectively. The association between ≥10,000 cells and CFU per mL with elevated biomarkers in symptomatic patients suggests that this lower threshold may be more suitable than 100,000 CFU/mL for diagnosing UTIs.

## 1. Introduction

In published guidelines, there is a pressing need for greater consensus regarding the minimum microbial threshold for diagnosing a urinary tract infection (UTI) [1,2,3,4,5,6,7,8,9]. The existing literature presents conflicting information on the commonly used diagnostic threshold of 100,000 colony-forming units (CFUs)/mL, with surprisingly scant and dated evidence to support it [1,2,3,4,5,6]. The confusion regarding a minimum threshold has led to uncertainty amongst clinicians, which can lead to increased use of empiric therapy or undertreatment of UTIs caused by lower microbial densities [10]. Ensuring accurate diagnosis of symptomatic patients with complicated urinary tract infections (cUTIs) is crucial because these patients often possess one or more risk factors that can lead to treatment failure, adverse clinical outcomes, and/or severe complications [11,12,13]. There is a gap in contemporary studies evaluating the correlation between rising microbial density and the presence or absence of a UTI [14].

Due to the limitations of standard urine culture (SUC), which make it a flawed gold standard test for diagnosis, it is important to identify cases that have a very high likelihood of being true UTIs in any study evaluating diagnostic tests and thresholds [15,16]. This study defined true UTI cases as specimens from symptomatic patients that had a clinical diagnosis in a specialist setting for UTI, had a positive identification of known uropathogens via multiplex polymerase chain reaction test (M-PCR) or SUC, and had elevated urine biomarkers that were documented to show inflammation of the urinary tract and have high specificity for UTI.

The uroepithelium and resident innate immune cells in the bladder quickly protect against microbial threats by first identifying microbial patterns and triggering an immune response that produces antimicrobial peptides and pro-inflammatory cytokines [17,18,19,20]. The biomarkers used in this study [neutrophil gelatinase-associated lipocalin (NGAL), interleukin-8 (IL-8), and interleukin-1-beta (IL-1β)] are essential components of the constitutive immune response in the urinary tract and have been studied in association with UTIs. Importantly, these three biomarkers have also demonstrated utility for the diagnosis of UTIs [21,22,23,24,25,26,27,28]. One of our previous studies discovered that NGAL, IL-8, and IL-1β have sensitivities of 82.6%, 91.2%, and 69.8% with specificities of 90.8%, 76.8%, and 96.9%, respectively [29]. A consensus of two or more of these biomarkers meeting the threshold of positivity yielded a sensitivity of 84.0% and a specificity of 91.2% [29].

Using these biomarkers plus a UTI diagnosis in a specialty setting to identify UTI cases in patients 60 years of age and older, we evaluated biomarker levels at different microbial densities using both M-PCR and SUC. The aim was to determine if there was a particular minimal microbial density threshold at which the biomarkers in these cases were significantly elevated, indicating that below this density, a UTI was unlikely. There is common agreement that no absolute minimum density threshold should be set, since some patients have been shown to have severe UTIs even at a very low density of 10^2^ CFU/mL [10]. However, it would be useful to assess if there is a threshold over which a significant majority of UTIs are present in symptomatic patients. In this study, we examine several different categories of microbial densities and their relationship to levels of inflammatory biomarkers in order to determine if a threshold significantly lower than 100,000 CFU/mL is more appropriate as a general benchmark to diagnose UTIs [30]. Evidence of a lower threshold would call into question the long-standing practice of considering lower densities as incidental findings, which has the potential to lead to underdiagnosis of patients showing symptoms of a UTI [31].

## 2. Materials and Methods

### 2.1. Study Design

This study utilized banked urine specimens from patients presenting at urology clinics in 39 U.S. states and assigned ICD-10-CM codes consistent with UTI, and compared urine inflammation biomarker results to microbial quantification results. These ICD-10-CM codes are assigned in the urology specialty setting based on the clinical presentation of the patient and are routinely transmitted to the lab with the diagnostic test order and urine specimen. Only specimens sent for UTI diagnostic testing that also had ICD-10-CM codes that indicated either a UTI or a UTI-related concern were selected for this study. There were 583 specimens included from consecutive eligible subjects which were collected between 17 January 2023 and 24 April 2023. Full inclusion and exclusion criteria are described in Appendix A.

Subjects’ de-identified urine samples were stored in a biorepository and evaluated at Pathnostics’ clinical laboratory. The Western Institutional Review Board deemed this remnant sample study to be exempt under 45 CFR § 46.104(d)(4), as the information is used in a manner such that the identity of the subject cannot be readily ascertained directly or via identifiers linked to the subjects, the subject is not contacted, and the investigator will not re-identify subjects. Urine samples from any previous IRB-approved clinical trials where the patient specifically opted out from research use of their remnant samples and corresponding de-identified data were excluded.

### 2.2. Specimen Handling

All urine specimens in this study were collected into a sterile cup via the midstream clean-catch method. Specimens were split and transferred into two Vacutainers (Becton Dickinson, Franklin Lakes, NJ, USA), one yellow-top tube for the P-AST assays and one gray-top tube containing boric acid for the M-PCR, SUC, and biomarker testing. Upon receipt, each urine sample was separated into 1 mL aliquots and placed in microcentrifuge tubes labeled with unique codes that do not contain patient identifiers. Labels were placed securely on each tube and scanned into software for biobanking and future tracking. The only data associated with each biobanked sample were the age and sex of the patient and any associated ICD-10-CM codes.

Aliquots immediately underwent testing via M-PCR and SUC. Aliquots for biomarker (NGAL, IL-8, and IL-1β) testing were centrifuged at 13,200 rpm for 2 min. The aspiration of the supernatant was transferred to a clean tube, labeled, and frozen at −80 °C +/−10 °C until ELISA testing.

### 2.3. Specimen Testing

Enzyme-linked immunosorbent assay (ELISA)- ELISA kits purchased from R&D Systems/Bio-Techne (Minneapolis, MN, USA), including human Lipocalin-2/NGAL Quantikine ELISA Kit (Catalog number SLCN20), human IL-8/CXCL8 Quantikine ELISA Kit (Catalog number S8000C), and human IL-1β/IL-1F2 Quantikine ELISA kit (Catalog number SLB50) were used. The assays measured the levels of NGAL [range 0.2–500 ng/mL], IL-8 [range 7.5–2000 pg/mL], and IL-1β [range 3.9–250 pg/mL] in the urine study specimens per the manufacturer’s instructions. An Infinite M Nano+ microplate reader (TECAN, Switzerland) measured absorbance at 450 nm and 540 nm, respectively. Frozen supernatants were thawed at room temperature before assaying.

The multiplex-polymerase chain reaction (M-PCR) and pooled antibiotic susceptibility testing (P-AST), M-PCR/P-AST assays (Guidance^®^ UTI, Pathnostics, Irvine, CA, USA) were used for susceptibility testing for 19 antibiotics, semi-quantitation of 27 pathogens, 3 bacterial groups, the ESBL phenotype, and the identification of 32 antibiotic-resistance genes. This test is intended to be used for the diagnosis of complicated, persistent, or recurrent UTIs and for UTIs in elevated-risk patients. The assay was performed as previously described [32]. First, the King Fisher/MagMAX™ automated DNA extraction instrument and the MagMAX™ DNA Multi-Sample Ultra Kit (Thermo Fisher, Carlsbad, CA, USA) extracted microbial DNA from the urine specimen according to the manufacturer’s instructions. The extracted DNA was used to identify and quantitate the specimen microbes. After combining a universal PCR master mix and the extracted DNA, amplification was completed using TaqMan technology in a Life Technologies 12 K Flex 112-format OpenArray System (Thermo Fisher Scientific, Wilmington, NC, USA). The inhibition PCR control used was *Bacillus atrophaeus*. Plasmids containing bacterial target DNA unique to each microbial species acted as positive controls. Duplicate specimen DNA samples were spotted on 112-format OpenArray chips (Thermo Fisher Scientific, Wilmington, NC, USA). The Pathnostics data analysis tool (Pathnostics, Irvine, CA, USA) sorted data, assessed data quality, summarized control sample data, identified positive assays, quantified bacterial load, and generated results. The results of the antibiotic resistance gene detection and the P-AST component of the test, which provides pooled phenotypic susceptibility results for 19 antibiotics, were not included in this analysis.

Quantitative M-PCR used probes and primers to detect the following microbes (23 bacteria species, 4 yeast species, and 3 bacterial groups): *Acinetobacter baumannii* (*A. baumannii*); *Actinotignum schaalii* (*A. schaalii*); *Aerococcus urinae* (*A. urinae*); *Alloscardovia omnicolens* (*A. omnicolens*)*; Candida albicans* (*C. albicans*); *Candida auris* (*C. auris*); *Candida glabrata* (*C. glabrata*); *Candida parapsilosis* (*C. parapsilosis*); *Citrobacter freundii* (*C. freundii*); *Citrobacter koseri* (*C. koseri*); *Corynebacterium riegelii* (*C. riegelii*)*; Enterococcus faecalis* (*E. faecalis*); *Enterococcus faecium* (*E. faecium*)*; Escherichia coli* (*E. coli*); *Gardnerella vaginalis* (*G. vaginalis*); *Klebsiella oxytoca* (*K. oxytoca*); *Klebsiella pneumoniae* (*K. pneumoniae*); *Morganella morganii* (*M. morganii*); *Mycoplasma hominis* (*M. hominis*); *Pantoea agglomerans* (*P. agglomerans*); *Proteus mirabilis* (*P. mirabilis*); *Providencia stuartii* (*P. stuartii*); *Pseudomonas aeruginosa* (*P. aeruginosa*); *Serratia marcescens* (*S. marcescens*); *Staphylococcus aureus* (*S. aureus*); *Streptococcus agalactiae* (*S. agalactiae*); *Ureaplasma urealyticum* (*U. urealyticum*); Coagulase Negative Staphylococci (CoNS), which includes *Staphylococcus epidermidis*, *Staphylococcus haemolyticus, Staphylococcus lugdunenesis,* and *Staphylococcus saprophyticus* (*S. saprophyticus*); the Enterobacter Group, which includes *Klebsiella aerogenes* (*K. aerogenes*) (*formally known as Enterobacter aerogenes*) and *Enterobacter cloacae* (*E. cloacae*); and Viridans Group Streptococci (VGS), which includes *Streptococcus anginosus*, *Streptococcus oralis*, and *Streptococcus pasteuranus*. Generated reports provided the name(s) of all yeasts detected at any level and all bacteria detected at a density range of <10,000, 10,000–49,999, 50,000–99,999, or ≥100,000 in cells/mL. The cells/mL quantitation was previously shown to correlate linearly, 1:1, with CFUs/mL as defined by SUC [33].

Standard urine culture (SUC), was performed as previously described [32]. Briefly, urine was vortexed, and samples of 1µL each were spread onto blood agar and colistin and nalidixic acid agar/MacConkey agar (CNA/MAC) plates, respectively, using a sterile plastic loop. All plates were incubated for 24 h at 35 °C under aerobic conditions. Plates with <10,000 CFUs/mL were reported as normal urogenital flora, and plates with growth ≥ 10,000 CFU/mL were used for colony counts (blood agar plates) and identification and quantitation of each morphologically distinct and separate colony (CNA/MAC plates). If ≥3 pathogens were present without a predominant species, results were reported as contaminated/mixed flora. Pathogen identification was confirmed via the VITEK 2 Compact System (bioMerieux, Durham, NC, USA) [32].

### 2.4. Statistical Analysis

Participant demographics and the ICD-10-CM code breakdown are described in the summary statistics table (e.g., mean and standard deviation (SD) for continuous variables, such as age, or count and percentage for categorical variables, such as sex and ICD-10-CM. The distribution of all the microorganisms listed above is provided with the number of each species detected and population percentages via M-PCR and by SUC. Summary statistics (*n*, median, mean) for the expression of the three biomarkers are provided for different subgroups: microbial density and detection method, either M-PCR or SUC. Each subgroup median value was compared to the “no microbes detected” group median using the Wilcoxon test. Previously published thresholds for biomarker positivity were used as cutoffs for the analysis: NGAL ≥ 38 ng/mL, IL-8 ≥ 20.6 pg/mL, and IL-1β ≥ 12.4 pg/mL [34,35]. This study defined biomarker consensus as any two or all three of the biomarkers positive at or above the cutoff levels. Individual biomarker results were evaluated with summary statistics (*n*, median, mean). Results were compared between different microbial density categories derived via both M-PCR and SUC. All hypothesis tests were two-sided, and a *p*-value of <0.05 was considered statistically significant. All data analyses were performed using R 4.2.2 (https://www.r-project.org/ (accessed on 11 January 2022)).

## 3. Results

### 3.1. Patient Demographics

We examined urine samples from a total of 583 individuals with a mean age of 76.6 years (standard deviation 8.85, median 76.3, range 60.0–99.7)). Of these, 68.3% (*n* = 398) were females, and 31.7% (*n* = 185) were males. Most symptomatic patient samples were submitted with an ICD-10-CM code (https://www.icd10data.com (accessed on 11 January 2022)) of N39.0 for “Urinary tract infection, site not specified” [81.8% (*n* = 534)]; see Table 1.

### 3.2. Bacterial and Yeast Identification

Out of the 583 urine samples obtained from patients with UTI symptoms, M-PCR did not detect any microbes in 117 samples, while SUC did not detect any microbes in 193 samples. In samples in which microorganisms were detected, M-PCR identified 883 microbes with densities of ≥10,000 cells/mL, while SUC identified 496 microorganisms at a threshold of ≥10,000 CFU (Appendix A).

### 3.3. Biomarker Detections

Specimen levels of NGAL, IL-8, and IL-1β biomarkers were measured via ELISA. These levels were plotted against four ranges of microbial densities detected via M-PCR and SUC: no microorganisms detected; <10,000, 10,000 to 99,999; and ≥100,000 cells or CFU per mL. Levels of each biomarker versus density ranges are shown in Figure 1, Figure 2 and Figure 3. The “no microorganisms detected” group was used for statistical comparisons, employing the Wilcoxon test to evaluate the differences between the biomarker level medians from each of the microbe density ranges. Table 2, Table 3 and Table 4 present the comprehensive summary of comparisons between each of the three biomarkers and density categories from both PCR and SUC.

In both the M-PCR and SUC methodologies, there was a consistent rise in the median levels of the three biomarkers as the microbial density increased. Compared to the group with no microbes detected, biomarkers showed significantly higher median levels, starting in the 10,000 to 99,999 cells/mL category for M-PCR (*p* = 0.0002, *p* < 0.0001, *p* < 0.0001) and SUC (*p* = 0.0104, *p* = 0.0014, *p* = 0.0064) for NGAL, IL-8, and IL-1β, respectively. The ≥100,000 cells/mL category also displayed a significantly higher median level of each biomarker according to both M-PCR (*p* < 0.0001 for all) and SUC (*p* < 0.0001 for all). There was no significant difference in the median levels between specimens with no microbes detected and specimens with microbes detected at <10,000 cells/mL for M-PCR (*p* = 0.2076, *p* = 0.7018, *p* = 0.86) or CFU/mL for SUC (*p* = 0.8056, *p* = 0.7767, *p* = 0.2083) for NGAL, IL-8, or IL-1β, respectively.

### 3.4. Biomarker Patterns: Microbial Density and Technique Differences

As observed in Table 2, Table 3, Table 4 and Figure 1, Figure 2, Figure 3, a consistent correlation exists between increased microbial density and elevated biomarker levels, regardless of the microbial detection method used. There was a significantly higher number of SUC specimens with negative results (*n* = 193) compared to M-PCR specimens (*n* = 117).

Median biomarker levels in the “no microorganisms detected” group were significantly higher for NGAL (*p* = 0.002), IL-8 (*p* = 0.009), and IL-1β (*p* = 0.001) when microbial detection was assessed via SUC compared to M-PCR. (Figure 4). There were also a higher number of specimens identified with microbial densities in the ≥100,000 cells/mL category when tested via M-PCR (*n* = 364) compared to SUC (*n* = 263). Several specimens from the “no microorganisms detected” SUC group had high biomarker levels and high microbial densities when tested via M-PCR. Out of the 193 samples that were negative based on SUC, 60 (31%) showed high levels of both microbes, ≥10,000 from M-PCR and NGAL (above threshold); 70 (36.2%) had elevated microbial levels, ≥10,000 from M-PCR, along with high levels of IL-8 (above threshold); and 47 (44.4%) had microbial levels ≥ 10,000 by M-PCR combined with biomarker levels that surpassed the threshold for IL-1β (above threshold).

### 3.5. Biomarker Consensus by Microbial Density

The assessment of each specimen for biomarker consensus (defined as two or more biomarkers above the established cutoffs) was conducted and organized based on microbial density and detection method (Figure 5). Specimens having microbial densities ≥ 10,000 cells/mL detected via either M-PCR or SUC exhibited over 50% biomarker consensus positivity with the percent of positivity increasing as microbial density rose. SUC-negative cases, in which no microbes were detected (*n* = 193), had biomarker consensus-positive results in 46% of cases, which was much higher than M-PCR negative cases (*n* = 117) with 29% being consensus-positive.

## 4. Discussion

In this study, we employed the current standard-of-care SUC method and the culture-independent M-PCR assay to identify and quantify microbes from patients with presumptive UTIs. Previous studies have demonstrated a 1:1 correlation for microbial quantitation in the linear range between SUC and M-PCR, allowing for direct comparisons between culture-dependent and culture-independent methods across various microbial densities [33]. Additionally, we assessed the immune response by measuring infection-associated biomarkers (NGAL, IL-8, and IL-1β) within the same urine specimens. This unique approach facilitated meaningful comparisons between SUC and M-PCR and provided direct associations between the presence and density of microorganisms against the immune response in clinically relevant specimens.

Traditionally, a microbial density of ≥100,000 cells/mL has been deemed diagnostically significant; however, recent clinical reviews and guidelines have proposed lower thresholds that are still clinically relevant [1,2,3,4,5,6]. In our study, we observed that lower microbial densities (≥10,000 cells/mL) detected via M-PCR and SUC in symptomatic subjects showed a notable increase in infection-associated biomarker levels. Additionally, for these subjects suspected of having a UTI, a cell density of ≥10,000 cells/mL from M-PCR and SUC strongly correlated with biomarker consensus. These findings suggest that a microbial detection threshold of 10,000 cells/mL could be an indicative criterion for diagnosing a UTI.

Using a threshold of >100,000 cells/mL as a criterion for initiating antimicrobial therapy carries significant clinical implications. One consequence is the potential for undertreatment of certain UTI patients, allowing microbes to proliferate, infiltrate host cells, develop biofilms, or acquire antibiotic resistance before treatment is administered [36,37]. Delaying treatment can lead to a progression of clinical severity, potentially necessitating higher antibiotic doses for extended durations or increasing the likelihood of complications, including recurrent infections and bacteremia [38].

Previous studies consistently show that M-PCR outperforms SUC in identifying and quantifying more causative uropathogens [32,39,40]. This is significant in that the SUC method favors the growth of gram-negative microbes such as *E. coli* but is less effective for non-*E. coli* microorganisms and fastidious microbes, which are increasingly recognized as common uropathogens [15,16,41,42,43,44,45]. Additionally, SUC is less likely to identify polymicrobial infections [32,39,40].

Building on these findings, our study also observed more specimens in which no microorganisms were detected (*n* = 193) in the SUC group compared to the M-PCR group (*n* = 117). Interestingly, the specimens with no microorganisms detected via SUC exhibited significantly higher levels of NGAL, IL-8, and IL-1β than those with no microbes detected via M-PCR. Furthermore, SUC specimens in which no organisms were detected demonstrated significantly higher (46%) consensus scores (two or more biomarkers above the established thresholds) when compared to M-PCR negative cases (29%).

The significantly higher biomarker levels in SUC-negative cases compared to M-PCR-negative cases have important implications. Firstly, the difference in biomarker levels underscores the potential variation in effectiveness between SUC and M-PCR as testing methods for detecting potential pathogens, suggesting that M-PCR may have higher sensitivity and specificity for detecting microbes that are causing a UTI, thereby enhancing diagnostic accuracy for symptomatic cases of UTI. Furthermore, SUC cases displaying low or no microbial densities alongside high biomarker levels suggest a potential failure in detecting the organisms causing the UTI, leading to these elevated levels. These findings challenge the sensitivity of SUC for the identification of uropathogens, raising questions about false negatives in culture-based testing. Consequently, the increase in biomarker levels between SUC-negative cases highlights the need to carefully consider the testing method employed when interpreting results and making treatment decisions, as SUC-negative cases may warrant closer monitoring.

During the analysis of urine samples from patients symptomatic of a UTI, a few specimens exhibited outlier data points characterized by lower inflammatory biomarker levels despite high microbial densities. These outliers with low inflammatory biomarker levels, even in symptomatic cases, could reflect scenarios where immune responses are compromised due to medications or underlying health conditions or instances of a resolving UTI [46,47,48,49,50]. Furthermore, it is worth noting that different microbial strains may vary in their ability to elicit an immune response, and the combined presence of multiple strains can also influence the observed biomarker levels. Future work could explore the thresholds specific to individual or combinations of microbes and their pathogenic densities in detecting urinary tract infections.

This study has several strengths. Firstly, a large number of samples were concurrently tested for levels of three different biomarkers and microbial detection via two different semi-quantitative microbial detection assays. This comprehensive approach allowed us to semi-quantitatively measure microbial density while simultaneously assessing the optimal density for diagnosing a urinary tract infection (UTI) and determining the accuracy of a negative diagnostic test result. The inclusion of multiple biomarkers and assays enhances the robustness of the findings and strengthens the overall conclusions.

However, there are some limitations to consider. One weakness of the study is that we were unable to measure the biomarker levels based on specific symptoms experienced by the patients. Understanding how the biomarkers correlate with different symptoms could provide valuable insights into the diagnostic and prognostic capabilities of the assays. Additionally, the number of cases in our study was not sufficient to measure biomarker levels across densities for specific microbial species. Further investigations with a larger sample size would enable a more comprehensive analysis of the relationship between microbial densities, biomarker levels, and specific microbial species.

Future studies could build upon these findings by assessing the biomarker levels across different species or combinations of species, allowing for a more targeted approach to diagnosing and managing UTIs. Additionally, exploring the association between biomarker levels and specific symptoms of UTIs would provide valuable clinical information for personalized treatment strategies. These potential avenues of research can further enhance our understanding of urinary tract infections and contribute to improved diagnostic accuracy and patient care.

## 5. Conclusions

In conclusion, our study shows that symptomatic subjects with UTIs exhibit a significant immune response at a microbial density threshold of ≥10,000 cells/mL, regardless of the detection method used. This suggests that a lower diagnostic microbial density threshold is clinically appropriate for UTI diagnosis and management, applicable to both the microbial identification and quantitation techniques used here. Additionally, our study emphasizes the need for closer monitoring when SUC yields a negative result, given the possibility of false negatives and their clinical implications.

## Figures and Tables

**Figure 1 diagnostics-13-02688-f001:**
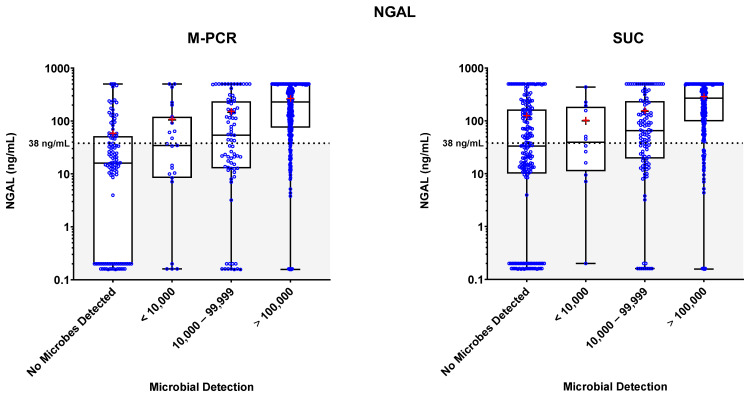
Box plots of NGAL biomarker levels in ng/mL are shown for different microbial density categories based on M-PCR quantitative data (**left**) and SUC quantitative data (**right**). Biomarker concentrations from individual urine specimens are indicated by open blue circles. Whiskers extend from the minimum to the maximum detected biomarker concentrations, biomarker medians for each microbial density category are marked with a horizontal line and used for statistical analysis, and biomarker means for each microbial density category are marked with a red “+”. The black dotted line marks the threshold for NGAL positivity (38 ng/mL), and shading indicates values below the threshold.

**Figure 2 diagnostics-13-02688-f002:**
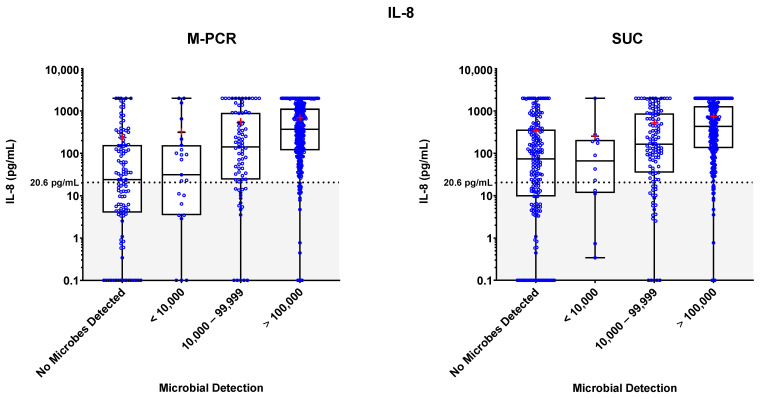
The box plots of IL-8 biomarker levels in pg/mL are displayed for different microbial categories based on M-PCR quantitative data (**left**) and SUC quantitative data (**right**). Biomarker concentrations from individual urine specimens are indicated by open blue circles. Whiskers extend from the minimum to the maximum biomarker detected concentration, biomarker medians for each microbial density category are marked with a horizontal line and used for the statistical analysis, and means for each microbial density category are marked with a red “+”. The black dotted line marks the threshold for IL-8 positivity (20.6 pg/mL), and shading indicates values below the threshold.

**Figure 3 diagnostics-13-02688-f003:**
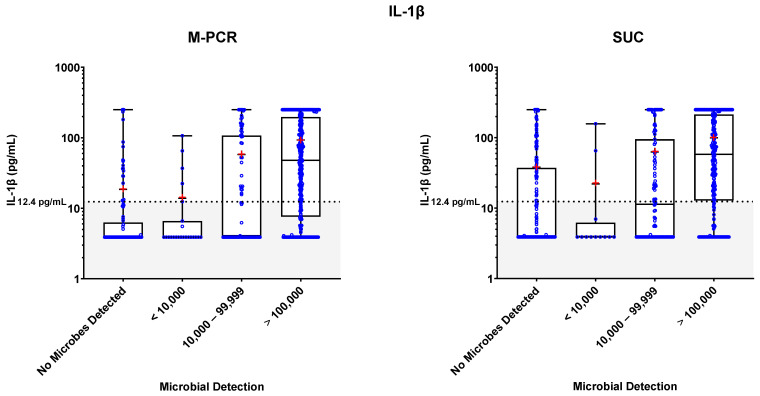
The box plots of IL-1β biomarker levels in pg/mL are displayed for different microbial categories based on M-PCR quantitative data (**left**) and SUC quantitative data (**right**). Biomarker concentrations from individual urine specimens are indicated by open blue circles. Whiskers extend from the minimum to the maximum biomarker concentration, biomarker medians for each microbial density category are marked with a horizontal line and used for statistical analysis, and means for each microbial density category are marked with a red “+”. The black dotted line marks the threshold for IL-1β positivity (12.4 pg/mL), and shading indicates values below the threshold.

**Figure 4 diagnostics-13-02688-f004:**
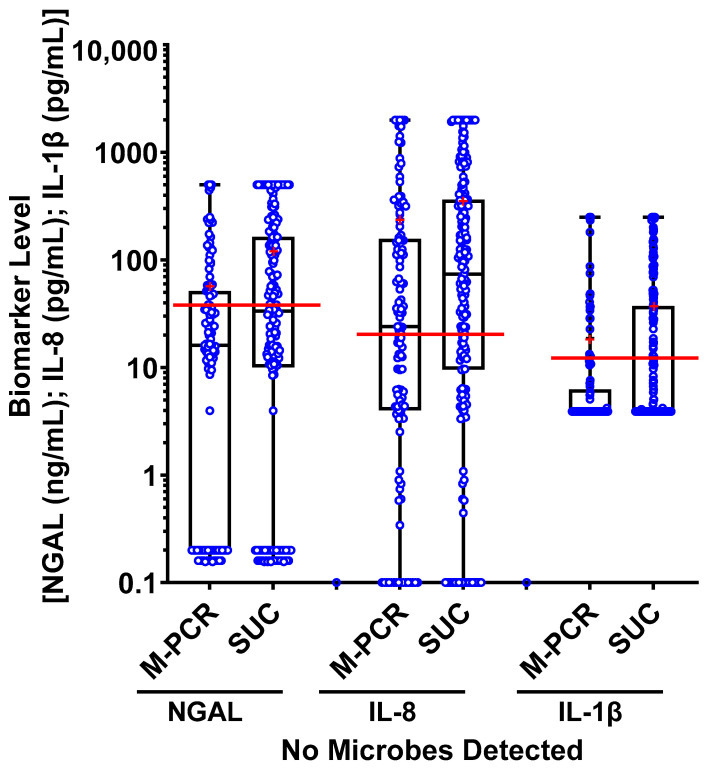
The box plots of NGAL (ng/mL), IL-8 (pg/mL), and IL-1β (pg/mL) biomarker levels from specimens in which no microbes were detected (*n* = 117 by M-PCR, *n* = 193 by SUC). Values from individual urine specimens are indicated by open blue circles. Whiskers extend from the minimum to the maximum detected value; means are indicated with a red “+”; medians for each microbial density category are marked with a horizontal line and used for statistical analysis.

**Figure 5 diagnostics-13-02688-f005:**
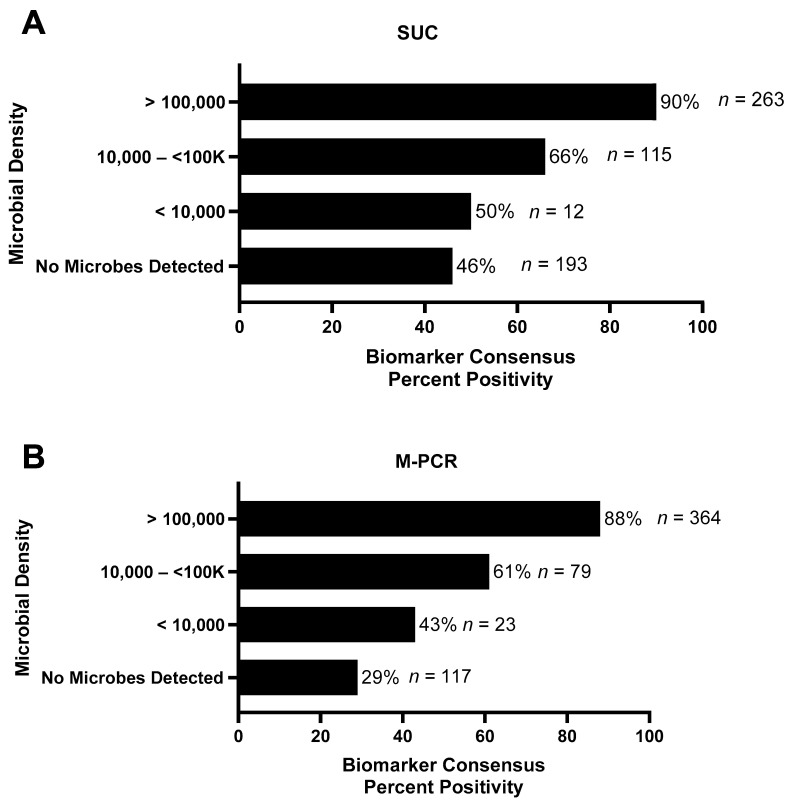
Biomarker consensus displayed for different microbial densities, and biomarker consensus positivity percentage plotted along the *x*-axis. Microbial density categories as detected via SUC (**A**) and M-PCR (**B**). The number of specimens in each microbial density category and the biomarker consensus percent positivity are labeled at the end of each bar.

**Table 1 diagnostics-13-02688-t001:** Top ICD-10-CM codes from the symptomatic study cohort.

ICD-10-CM Code	Code Description	Frequency	Percent
N39.0	Urinary tract infection, site not specified	534	81.8
R30.0	Dysuria	43	6.6
R31.0	Gross hematuria	25	3.8
Z87.440	Personal history of diseases of urinary system	5	0.8
R31.9	Hematuria, unspecified	4	0.6
R82.998	Other abnormal findings in urine	4	0.6
Others	-	38	12.4

Some patients had more than one ICD-10-CM code associated with their case. The five most prevalent codes are listed individually, and all remaining codes are grouped together as “other”.

**Table 2 diagnostics-13-02688-t002:** Summary Descriptive Statistics of NGAL Biomarker Levels Stratified by Microbial Density as Detected by M-PCR and SUC.

NGAL	M-PCR				SUC			
Microbial Density Category	*n* (% of Total)	Median	Mean	*p*-Value (Relative to Symptomatic no Microbes Detected)	*n* (% of Total)	Median	Mean	*p*-Value (Relative to Symptomatic no Microbes Detected) *
No Microbes Detected	117 (20.0%)	16.05	56.68	-	193 (33.1%)	33.49	120.71	-
<10,000	23 (3.9%)	34.55	105.03	0.2076	12 (2.1%)	39.44	100.12	0.8056
10,000–99,999	79 (13.6%)	53.78	145.3	0.0002	115 (19.7%)	65.49	152.44	0.0104
≥100,000	364 (62.4%)	228.87	260.27	<0.0001	263 (45.1%)	268.81	278.46	<0.0001

* *p*-values calculated based on the medians.

**Table 3 diagnostics-13-02688-t003:** Summary Descriptive statistics of IL-8 Biomarker Levels Stratified by Microbial Density as Detected by M-PCR and SUC.

IL-8	M-PCR				SUC			
Microbial Density Category	*n* (% of Total)	Median	Mean	*p*-Value (Relative to Symptomatic no Microbes Detected)	*n* (% of Total)	Median	Mean	*p*-Value (Relative to Symptomatic no Microbes Detected) *
No Microbes Detected	117 (20.0%)	23.95	236.57	-	193 (33.1%)	73.84	354.64	-
<10,000	23 (3.9%)	31.22	312.06	0.7018	12 (2.1%)	66.30	253.22	0.7767
10,000–99,999	79 (13.6%)	141.84	531.96	<0.0001	115 (19.7%)	164.57	523.82	0.0014
≥100,000	364 (62.4%)	371.12	685.69	<0.0001	263 (45.1%)	431.6	740.5	<0.0001

* *p*-values calculated based on the medians.

**Table 4 diagnostics-13-02688-t004:** Summary Descriptive statistics of IL-1β Biomarker Levels Stratified by Microbial Density as Detected by M-PCR and SUC.

IL-1β	M-PCR				SUC			
Microbial Density Category	*n* (% of Total)	Median	Mean	*p*-Value (Relative to Symptomatic no Microbes Detected)	*n* (% of Total)	Median	Mean	*p*-Value (Relative to Symptomatic no Microbes Detected) *
No Microbes Detected	117 (20.0%)	3.9	18.57	-	193 (33.1%)	3.9	38.17	-
<10,000	23 (3.9%)	3.9	13.86	0.86	12 (2.1%)	3.9	22.09	0.2083
10,000–99,999	79 (13.6%)	4.08	57.95	<0.0001	115 (19.7%)	11.38	62.75	0.0064
≥100,000	364 (62.4%)	47.85	93.4	<0.0001	263 (45.1%)	57.88	99.69	<0.0001

* *p*-values calculated based on the medians.

## Data Availability

The data presented in this study are available on request from the corresponding author. The data are not publicly available due to privacy concerns.

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
