# Peer review of "Elevated UTI Biomarkers in Symptomatic Patients with Urine Microbial Densities of 10,000 CFU/mL Indicate a Lower Threshold for Diagnosing UTIs"

_diagnostics, 2023, doi:10.3390/diagnostics13162688_

Round 1

Reviewer 1 Report

Dear authors

In the current manuscript, you aimed to identify changes in immune response at different microbial densities and determine an optimal threshold for diagnosing symptomatic UTIs. In symptomatic subjects, you found an association between microbial densities ≥ 10,000 cells/mL and NGAL, IL-8, and IL-1β levels. You suggested that this low threshold (10,000 cells) may be more suitable for diagnosing UTIs. Bottom of Form

Top of Form

 However, there are many comments on this manuscript:

The title:

·       It is very long and includes numbers. This needs rephrasing.

Abstract:

·       The partitions of the abstract are missing and the study design is unclear.

Introduction:

·       The hypothesis is not clear.

·       Introduction needs more clarification for the markers used.

·       Many abbreviations are mentioned without their complete names (A complete name should be mentioned before the abbrev. for the first time).

·       The authors in their previous study have proven that the studied biomarkers (NGAL, IL-8, and IL-1β) may be associated with complicated UTIs. How can they be used as early biomarkers for diagnosis? They should be sued for prognosis.

·       Lines between 52-54, the references cited from 17 to 31, These references discuss many issues related to UTI and biomarkers as diagnosis, recurrence, prog, and comparison between biomarkers, not only to prove the relation with cUTI.

 Materials and Methods:

·       The study design is unclear.

·       Inclusion and exclusion criteria are not present

·       Why patients are not categorized.

·       What are the symptoms used to judge symptomatic UTI.

·       Ethical statement and informed consent is not mentioned.

·       Multiplex- Polymerase Chain Reaction (M-PCR) and Pooled Antibiotic Susceptibility Testing (P-AST)- The M-PCR/P-AST assay is very long, just brief is better.

Results:

·       Mean and SD of age is better to be mentioned.

·       Author mentioned that 117 samples did not include any organism. This indicates that there are other reasons for the manifestation not UTI. These samples should be excluded from the other results.

·       No correlation was mentioned between each symptom (as the manuscript depends mainly on separate symptoms as mentioned in the beginning of the results) and biomarker or CFU.

Discussion:

·       More studies are needed and more clarification about the relation between these biomarkers and UTI is required.

Conclusion:

·       It is not consistent with the results as most of the studied populations have CFU > 10000.

References:

·       Some references are very old (1992, 1979, 1986, 1997, 2000, ….etc ).

·       Some ref. are not related to the topic and may be for self-citation as 37, 38, and 40.

 Other comments are in the attached PDF.

Best regards,

Author Response

Note: During the revisions process, we discovered a transcription error in Tables 2-4 for the median and mean values for the SUC >100,000 CFUs/mL microbial density category. This error did not affect the figures, results, or conclusions text, and the tables have been corrected.

Reviewer 1

Dear authors

In the current manuscript, you aimed to identify changes in immune response at different microbial densities and determine an optimal threshold for diagnosing symptomatic UTIs. In symptomatic subjects, you found an association between microbial densities ≥ 10,000 cells/mL and NGAL, IL-8, and IL-1β levels. You suggested that this low threshold (10,000 cells) may be more suitable for diagnosing UTIs.

 However, there are many comments on this manuscript:

The title:

  • It is very long and includes numbers. This needs rephrasing.

Response: Thank you for your suggestion. We have shortened and rephrased the title to: “Elevated UTI Biomarkers in Symptomatic Patients with Urine Microbial Densities of 10,000 CFU/mL Indicate a Lower Threshold for Diagnosing UTIs”.

Abstract:

  • The partitions of the abstract are missing and the study design is unclear.

Response: Thank you for your feedback. The Abstract lacks subheadings in alignment with the journal’s “Instructions for Authors” (https://www.mdpi.com/journal/diagnostics/instructions#:~:text=Abstract%3A%20The,the%20main%20conclusions.) “Abstract: The abstract should be a total of about 200 words maximum. The abstract should be a single paragraph and should follow the style of structured abstracts, but without headings: 1) Background: Place the question addressed in a broad context and highlight the purpose of the study; 2) Methods: Describe briefly the main methods or treatments applied. Include any relevant preregistration numbers, and species and strains of any animals used; 3) Results: Summarize the article's main findings; and 4) Conclusion: Indicate the main conclusions or interpretations. The abstract should be an objective representation of the article: it must not contain results which are not presented and substantiated in the main text and should not exaggerate the main conclusions.”

Additional details of the study design have been added to the abstract (see lines 15-30): “The literature lacks consensus on the minimum microbial density required for diagnosing urinary tract infections (UTIs). This study categorized the microbial densities of urine specimens from symptomatic UTI patients > 60 and correlated them to detected levels of immune response biomarkers neutrophil gelatinase-associated lipocalin (NGAL), interleukin-8 (IL-8), and interleukin-1-beta (IL-1β). The objective was identifying the microbial densities associated with significant elevation of these biomarkers in order to determine an optimal threshold for diagnosing symptomatic UTIs. Biobanked midstream voided urine samples were analyzed for microbial identification and quantification using standard urine culture (SUC) and multiplex-polymerase chain reaction (M-PCR), while NGAL, IL-8, and IL-1β levels were measured with enzyme-linked immunosorbent assay (ELISA). NGAL, IL-8, and IL-1β levels were all significantly elevated at microbial densities ≥ 10,000 cells/mL by M-PCR (p < 0.0069) or equivalent colony-forming units (CFUs)/mL by SUC (p < 0.0104) compared to samples with no detectable microbes.  With both PCR and SUC, a consensus of two or more elevated biomarkers correlated well with microbial densities > 10,000 cells/mL or CFU/mL respectively. The association between ≥ 10,000 cells and CFU per mL with elevated biomarkers in symptomatic patients suggests that this lower threshold may be more suitable than 100,000 CFU/mL for diagnosing UTIs.”

Introduction:

  • The hypothesis is not clear.

Response: Thank you for your feedback. We have added clarification of our hypothesis in lines 74-77: “In this study we examine several different categories of microbial densities and their relationship to the levels of inflammatory biomarkers, in order to determine if a significantly lower threshold from 100,000 CFU/mL is more appropriate as a general bench-mark to diagnose UTIs. [30]”

  • Introduction needs more clarification for the markers used.

Response: We appreciate your request for clarification. We have rephrased a portion of the introduction to clarify how the existing literature has demonstrated that these biomarkers have utility diagnostic indications related to UTI. See lines 61-62: “Importantly, these three biomarkers have also demonstrated utility for the diagnosis of UTI. [21–28] “

  • Many abbreviations are mentioned without their complete names (A complete name should be mentioned before the abbrev. for the first time).

Response: Thank you for pointing out the lack of complete names with our abbreviations and acronyms. These have been addressed in the Abstract and Introduction. (See lines 18-19, 22-25, 38, 47, and 52)

  • The authors in their previous study have proven that the studied biomarkers (NGAL, IL-8, and IL-1β) may be associated with complicated UTIs. How can they be used as early biomarkers for diagnosis? They should be used for prognosis.

Response: Thank you for your question. We have added clarification with cited references that demonstrate that these biomarkers have utility for UTI diagnosis. (see responses above and below as well as manuscript lines 61-620)

  • Lines between 52-54, the references cited from 17 to 31, These references discuss many issues related to UTI and biomarkers such as diagnosis, recurrence, prognosis, and comparison between biomarkers, not only to prove the relation with cUTI.

Response: Thank you for your feedback. We have rephrased a portion of the introduction to clarify how the existing literature has demonstrated that these biomarkers have utility in diagnostic indications related to UTI. (see previous response above and manuscript lines 61-62). As clarified in the manuscript text, references 21-28 specifically relate to the use of these three biomarkers in UTI diagnosis.

  1. Horváth, J.; Wullt, B.; Naber, K.G.; Köves, B. Biomarkers in Urinary Tract Infections – Which Ones Are Suitable for Diagnostics and Follow-Up? Gms Infect Dis 8, Doc24, doi:10.3205/id000068.
  2. Martino, F.K.; Novara, G. Asymptomatic Bacteriuria or Urinary Tract Infection? New and Old Bi-omarkers. Int J Transl Medicine 2022, 2, 52–65, doi:10.3390/ijtm2010006.
  3. Edwards, G.; Seeley, A.; Carter, A.; Smith, M.P.; Cross, E.L.; Hughes, K.; Bruel, A.V. den; Llewelyn, M.J.; Verbakel, J.Y.; Hayward, G. What Is the Diagnostic Accuracy of Novel Urine Biomarkers for Urinary Tract Infection? Biomark Insights 2023, 18, 11772719221144460, doi:10.1177/11772719221144459.
  4. Nanda, N.; Juthani-Mehta, M. Novel Biomarkers for the Diagnosis of Urinary Tract Infection–-A Sys-tematic Review. Biomark Insights 2009, 4, BMI.S3155, doi:10.4137/bmi.s3155.
  5. Shaikh, N.; Martin, J.M.; Hoberman, A.; Skae, M.; Milkovich, L.; McElheny, C.; Hickey, R.W.; Gabri-el, L.V.; Kearney, D.H.; Majd, M.; et al. Biomarkers That Differentiate False Positive Urinalyses from True Urinary Tract Infection. Pediatr Nephrol 2020, 35, 321–329, doi:10.1007/s00467-019-04403-7.
  6. Rodhe, N.; Löfgren, S.; primary …, S.-J. of Cytokines in Urine in Elderly Subjects with Acute Cystitis and Asymptomatic Bacteriuria. 2009, doi:10.1080/02813430902757634.
  7. Masajtis-Zagajewska, A.; Nowicki, M. New Markers of Urinary Tract Infection. Clin Chim Acta 2017, 471, 286–291, doi:10.1016/j.cca.2017.06.003.
  8. Yilmaz, A.; Sevketoglu, E.; Gedikbasi, A.; Karyagar, S.; Kiyak, A.; Mulazimoglu, M.; Aydogan, G.; Ozpacaci, T.; Hatipoglu, S. Early Prediction of Urinary Tract Infection with Urinary Neutrophil Gelatinase Associated Lipocalin. Pediatr Nephrol 2009, 24, 2387, doi:10.1007/s00467-009-1279-6.32. Yim, H.E.; Yim, H.; Bae, E.S.; Woo, S.U.; Yoo, K.H. Predictive Value of Urinary and Serum Biomarkers in Young Children with Febrile Urinary Tract Infections. Pediatr Nephrol 2014, 29, 2181–2189, doi:10.1007/s00467-014-2845-0.
  9. Hosman, I.S.; Roić, A.C.; Lamot, L. A Systematic Review of the (Un)Known Host Immune Response Biomarkers for Predicting Recurrence of Urinary Tract Infection. Frontiers Medicine 2022, 9, 931717, doi:10.3389/fmed.2022.931717.

 Materials and Methods:

  • The study design is unclear.

Response: We appreciate your feedback. Expanded details have been added to our methods section. See lines 83-92: “This study utilized banked urine specimens of patients presenting at urology clin-ics in 39 U.S. states assigned ICD-10-CM codes consistent with UTI, and compared urine inflammation biomarker results to microbial quantification results. These ICD-10 codes were assigned in the urology specialty setting based on the clinical presentation of the patient and are routinely transmitted to the lab with the diagnostic test order and urine specimen.  Only specimens sent for UTI diagnostic testing that also had ICD-10 codes that were either UTI or UTI-related were selected for this study.  There were 583 specimens included from consecutive eligible subjects which were collected between 01/17/2023 and 04/24/2023. Full inclusion and exclusion criteria are described in Supple-mental Table S1.”

Lines 98-100: “Urine samples from any previous IRB-approved clinical trials where the patient spe-cifically opted out from research use of their remnant samples and corresponding de-identified data were excluded.”

And Lines 107-110: “labeled with unique codes that do not contain patient identifiers. Labels were placed securely on each tube and scanned into software for biobanking and future tracking. The only data associated with each biobanked sample were the age and sex of the patient and any associated ICD-10-CM codes.”

  • Inclusion and exclusion criteria are not present.

Response: Thank you for pointing out the absence of these criteria. The full inclusion and exclusion criteria have been added as Supplemental Table S1.

Supplementary Table S1. Inclusion and Exclusion Criteria.

Inclusion

Exclusion

At least 60 years of age

Failure to meet all inclusion criteria

Male or Female sex
(no predetermined quotas or ratios for participation)

Presenting to a urologist or urogynecologist in an outpatient setting

Requires microbial testing according to clinician judgment

Sample ordered with an ICD-10-CM code associated with UTI

Sample contained enough urine to conduct M-PCR/P-AST, SUC, and biomarker assays on the same sample

  • Why patients are not categorized.

Thank you for your question. The aim of the study was to examine what threshold cell density was associated with increased inflammatory biomarker levels for patients routinely diagnosed as UTIs in urology offices and for which uropathogens were identified.  It was not meant or powered to sub-categorize these patients further, which may be a useful future study to do.  The use of de-identified biobanked samples used in this study was sufficient for the aim of the study, and included only information on age and sex of the patient, the associated ICD-10-CM code (which was used to justify the diagnostic testing), the type of healthcare provider requesting a test, the location for which the urine was collected, and the results of the microbial identification and biomarker testing. Additional details of the study design have been added to the methods (see lines 83-110).

  • What are the symptoms used to judge symptomatic UTI.

Response: Thank you for your question. The determinations of UTI were made within a specialty urology setting by the healthcare providers with expertise in this disease. These urologic specialists assigned the ICD-10-CM codes to the patients based on the patients' clinical presentations, which could include typical UTI symptoms, such as urinary frequency, urinary urgency, dysuria, and suprapubic pain, Dipstick Urinalysis results, or other details. A clarification of this point has been added to the manuscript. See lines 85-88: “These ICD-10-CM codes were assigned in the urology specialty setting based on the clinical presentation of the patient and are routinely transmitted to the lab with the diagnostic test order and urine specimen.”

  • Ethical statement and informed consent is not mentioned.

Response: Thank you for your concern. The ethical statement and informed consent are addressed both in the methods section (lines 94-98): “The Western Institutional Review Board exempted the study from review, as the urine specimens were obtained from a biobank repository,” and in the declarations at the end of the manuscript (lines 405-407): “Institutional Review Board Statement: The study was conducted in accordance with the Declaration of Helsinki. Ethical review and approval were waived for the symptomatic cohort in this study due to the use of deidentified samples from a biobank repository.”

Informed Consent Statement: Patient consent was waived for the symptomatic cohort in this study due to the use of deidentified samples from a biobank repository.

  • Multiplex- Polymerase Chain Reaction (M-PCR) and Pooled Antibiotic Susceptibility Testing (P-AST)- The M-PCR/P-AST assay is very long, just brief is better.

Response: We appreciate your recommendation for brevity. The test used in this study is a proprietary test (Guidance® UTI, Pathnostics, Irvine, CA) which consists of two major components: Multiplex polymerase chain reaction (M-PCR) and pooled antibiotic susceptibility testing (P-AST). The authors felt that it was important to clarify the components of the test according to their full names. This was only written in full one time (as a heading in the methods section), and the authors don’t feel it would be appropriate to leave out of the Methods the technology used.

Results:

  • Mean and SD of age is better to be mentioned.

Response: We appreciate your suggestion and have included the mean age with standard deviation in lines 196-197: “We examined urine samples from a total of 583 individuals with a mean age of 76.6 years (standard deviation 8.85, median 76.3, range 60.0 – 99.7).”

  • Author mentioned that 117 samples did not include any organism. This indicates that there are other reasons for the manifestation not UTI. These samples should be excluded from the other results.

Response: Thank you for your recommendation. This study aimed to determine a microbial threshold with diagnostic utility in the routine diagnostic workup performed for suspected UTIs in the urology setting. The samples used in this study are representative of this routine diagnostic workup. Therefore, the samples included in this analysis contain a mix of cases positive and negative for microorganisms, as should be expected from samples sent for diagnostic testing for UTIs (Refs 1-3). Cases with UTI-associated symptoms and a presumptive UTI diagnosis that test negative for microorganisms may represent alternative diagnoses such as overactive bladder or interstitial cystitis. While microorganism-negative samples are considered their own subgroup in our analyses, these samples were intentionally included in order to realistically represent the variety of samples that undergo microbial identification testing during diagnostic workups for presumed UTI.

  1. Heytens S, De Sutter A, Coorevits L, et al. Women with symptoms of a urinary tract infection but a negative urine culture: PCR-based quantification of Escherichia coli suggests infection in most cases. Clin Microbiol Infect. 2017;23(9):647-652. doi:10.1016/j.cmi.2017.04.004
  2. Naber KG, Schito G, Botto H, Palou J, Mazzei T. Surveillance study in Europe and Brazil on clinical aspects and Antimicrobial Resistance Epidemiology in Females with Cystitis (ARESC): implications for empiric therapy. Eur Urol. 2008;54(5):1164-1175. doi:10.1016/j.eururo.2008.05.010
  3. Barnes HC, Wolff B, Abdul-Rahim O, et al (2021) A Randomized Clinical Trial of Standard versus Expanded Cultures to Diagnose Urinary Tract Infections in Women. J Urology 206:1212–1221. https://doi.org/10.1097/ju.0000000000001949
  • No correlation was mentioned between each symptom (as the manuscript depends mainly on separate symptoms as mentioned in the beginning of the results) and biomarker or CFU.

Response: As described above in more detail as a response to a previous question, the aim of the study was to examine what threshold cell density was associated with increased inflammatory biomarker levels for patients routinely diagnosed as UTIs in urology offices and for which uropathogens were identified.  It was not meant or powered to sub-categorize these patients further, which may be a useful future study to do.

Discussion:

  • More studies are needed and more clarification about the relation between these biomarkers and UTI is required.

Response: We appreciate your feedback. We have rephrased a portion of the introduction to clarify how the existing literature has demonstrated that these biomarkers have utility in both diagnostic and prognostic indications related to UTI, with specific references to the diagnostic use called out. (see lines 58-62) The authors agree that additional studies on the application of these biomarkers would be informative and this remains a potential future direction for exploration.

Conclusion:

  • It is not consistent with the results as most of the studied populations have CFU > 10000.

Response: Thank you for your comment. It is true that the number of specimens in our study with microbes detected at a density < 10,000 cells/mL by M-PCR (23) or CFUs/mL by SUC (12) accounting for 5% or less of the overall 583 specimens. However, this number was sufficiently powered for statistical comparisons (see tables 2 – 4). These statistical comparisons are consistent with the conclusion that samples with microbial densities > 10,000 cells/mL or CFUs/mL, but NOT samples with < 10,000 cells/mL or CFUs/mL exhibited statistically significant increases in infection-associated biomarkers.

References:

  • Some references are very old (1992, 1979, 1986, 1997, 2000, ….etc ).

Response: In a recently submitted review article, collaborators examined the primary evidence used to justify diagnostic microbial thresholds for UTI and found that almost two-thirds (64%) of the 36 guidelines and 144 recommendations were published between 2016 and 2023, yet the evidence cited to justify the recommendations relied heavily on 23 research articles published between 1956 and 2013; thus there is limited new research to site. In fact, the most frequently cited justification reference was published in 1982:

Stamm WE, Counts GW, Running KR, Fihn S, Turck M, Holmes KK. Diagnosis of Coliform Infection in Acutely Dysuric Women. New Engl J Medicine. 1982;307(8):463-468. doi:10.1056/nejm198208193070802

This highlights a lack of contemporary research in the area of UTI diagnosis and appropriate treatment thresholds. The sparsity of recent literature necessitates the reliance on more dated references and also underscores the importance of studies, like ours, which aim to provide more current and comprehensive evidence to justify microbial density thresholds used in UTI diagnosis.

  • Some ref. are not related to the topic and may be for self-citation as 37, 38, and 40.

Response: We appreciate your point. Two extraneous references have been removed. However, one reference was kept as the M-PCR and SUC methods reference. (see lines 131 and 168)

Other comments are in the attached PDF:

  • 01/17/2023 and 04/24/2023 study duration is very short

Response: We appreciate your concern. Previous studies (ref 1,2, and 3) of the three biomarkers on the diagnosis of UTI, each included approximately 200 subjects. Between 01/17/2023 and 04/24/2023, our study was able to acquire 583 unique patient specimens, more than double the sample size of the other studies. Therefore, we stopped the enrollment on 04/25/23, and started the study analysis.

References

  1. Gadalla AAH, Friberg IM, Kift-Morgan A, Zhang J, Eberl M, Topley N, et al. Identification of clinical and urine biomarkers for uncomplicated urinary tract infection using machine learning algorithms. Sci Rep-uk. 2019;9(1):19694. doi:10.1038/s41598-019-55523-x
  2. Tyagi P, Tyagi V, Qu X, Chuang YC, Kuo HC, Chancellor M. Elevated CXC chemokines in urine noninvasively discriminate OAB from UTI. Am J Physiol-renal. 2016;311(3):F548-F554. doi:10.1152/ajprenal.00213.2016
  3. Steigedal M, Marstad A, Haug M, Damås JK, Strong RK, Roberts PL, et al. Lipocalin 2 Imparts Selective Pressure on Bacterial Growth in the Bladder and Is Elevated in Women with Urinary Tract Infection. J Immunol. 2014;193(12):6081-6089. doi:10.4049/jimmunol.1401528

Reviewer 2 Report

The authors have made an interesting attempt at “UTI Biomarkers in Symptomatic Patients are Significantly Elevated at Microbial Densities of 10,000 cells or CFUs per mL Indicating a Lower Microbial Density Threshold for Diagnosing UTIs.” The manuscript is interesting; however, the authors need to justify the scientific writing manuscript. Some of the general comments are provided below:

1.     How were the 583 patients selected for inclusion in the study? Was there any selection bias?

2.     What criteria were used to determine the clinical presentations and ICD-10-CM diagnostic codes consistent with UTI?

3.     How was the de-identification of the urine samples ensured to protect patient privacy and confidentiality?

4.     What percentage of the urine samples did not detect any microbes using M-PCR and SUC? Are there any potential reasons for the negative results?

5.     Were there any significant differences between M-PCR and SUC in terms of their ability to detect microbes in urine samples?

6.     Did the rise in biomarker levels correlate with increasing microbial density consistently for both M-PCR and SUC methodologies?

7.     What is the significance of the significantly higher number of SUC specimens with negative results compared to M-PCR? Are there any potential reasons for this discrepancy?

8.     What are the implications of the significantly higher biomarker levels in the "no microorganisms detected" group when microbial detection was assessed by the SUC method compared to M-PCR? How do these findings impact the interpretation of the results?

9.     How were the cutoffs for biomarker consensus established? Were they based on previous studies, clinical guidelines, or other criteria?

10. What are the clinical implications of the study findings regarding the optimal microbial density threshold for diagnosing urinary tract infections? How might these findings impact current diagnostic practices?

11. How do the limitations of the study, such as the inability to measure biomarker levels based on specific symptoms and the sample size, affect the generalizability and reliability of the results?

12. Based on the conclusions of the study, what are the key takeaways for healthcare professionals and researchers in the field of urinary tract infections? What are the suggested areas for future research and clinical implementation?

Author Response

Reviewer 2

Comments and Suggestions for Authors

The authors have made an interesting attempt at “UTI Biomarkers in Symptomatic Patients are Significantly Elevated at Microbial Densities of 10,000 cells or CFUs per mL Indicating a Lower Microbial Density Threshold for Diagnosing UTIs.” The manuscript is interesting; however, the authors need to justify the scientific writing manuscript. Some of the general comments are provided below:

Note: During the revisions process, we discovered a transcription error in Tables 2-4 for the median and mean values for the SUC >100,000 CFUs/mL microbial density category. This error did not affect the figures, results, or conclusions text; the tables have been corrected.

  1. How were the 583 patients selected for inclusion in the study? Was there any selection bias?

Response: Thank you for your concern. We recruited subjects presenting at urology clinics in 39 U.S. states with clinical presentations and ICD-10-CM diagnostic codes consistent with UTI and aimed to compare their urine inflammation biomarker levels (IL-8, IL-1ß, and N-GAL) and microbial quantification results.  Previous studies (ref 1,2, and 3) of the three biomarkers on the diagnosis of UTI, each included approximately 200 subjects. In order to prevent delays in sample handling associated with weekends, we only recruited subjects on three weekdays, Tuesday through Thursday. From 01/17/2023 and 04/24/2023, on these target recruiting weekdays, our study was able to acquire 583 unique patient specimens, more than double the sample size of the other studies. Therefore, we stopped the enrollment on 04/25/23, and started the study analysis. These subjects were consecutive cases, biobanked on these weekdays, which satisfied all inclusion criteria. Therefore, they were randomly selected, and thus were not associated with selection bias. A clarification of this point has been added to the manuscript. See lines 83-92: ”This study utilized banked urine specimens of patients presenting at urology clinics in 39 U.S. states assigned ICD-10-CM codes consistent with UTI, and compared urine inflammation biomarker results to microbial quantification results. These ICD-10-CM codes were assigned in the urology specialty setting based on the clinical presentation of the patient and are routinely transmitted to the lab with the diagnostic test order and urine specimen.  Only specimens sent for UTI diagnostic testing that also had ICD-10-CM codes that were either UTI or UTI-related were selected for this study.  There were 583 specimens included from consecutive eligible subjects which were collected between 01/17/2023 and 04/24/2023. Full inclusion and exclusion criteria are described in Supplemental Table S1.”

  1. What criteria were used to determine the clinical presentations and ICD-10-CM diagnostic codes consistent with UTI?

Response: We appreciate your question. The determinations were made by the healthcare providers with urologic experience and expertise. These urologic specialists assigned the ICD-10-CM codes to the patients based on the patient’s clinical presentations, which could include typical UTI symptoms defined such as urinary frequency, urinary urgency, dysuria, and suprapubic pain, Dipstick Urinalysis results, or other details. A clarification of this point has been added to the manuscript. See lines 85-88: “These ICD-10-CM codes were assigned in the urology specialty setting based on the clinical presentation of the patient and are routinely transmitted to the lab with the diagnostic test order and urine specimen.”

  1. How was the de-identification of the urine samples ensured to protect patient privacy and confidentiality?

Response: Thank you for your concern. The samples in the biorepository were de-identified to protect patient’s privacy and confidentiality, in accordance with the submission for IRB waiver to Western IRB, which was waived. The only data associated with each biobanked sample were the age and sex of the patient and any associated ICD-10-CM codes. No other information, such as patients’ names, medical record numbers, or addresses, were associated with the samples when received at the testing laboratory.

  1. What percentage of the urine samples did not detect any microbes using M-PCR and SUC? Are there any potential reasons for the negative results?

Response: Thank you for your questions. There were 117 (117/583, 20.1%) and 193 (193/583, 33.1%) urine samples that were not detected with any microbes using M-PCR and SUC, respectively (Table 2 of the manuscript). There was a higher percentage of SUC specimens with negative results compared to M-PCR. As described in the discussion of our manuscript (see lines 327-335), this observation is consistent with previous studies showing that M-PCR outperforms SUC in identifying and quantifying more causative uropathogens.(Refs 1-3) This is significant in that the SUC method favors the growth of gram-negative microbes like E. coli but is less effective for non-E. coli microorganisms and fastidious microbes, which are increasingly recognized as common uropathogens.(Refs 5-11) Additionally, SUC is less likely to identify polymicrobial infections. (Refs 1-3)

The growing body of literature on the urinary microbiome has examined the possible implications of dysbiosis or perturbations in the microbiome in conditions associated with chronic lower urinary tract symptoms (LUTS). (Ref 12) This study aimed to determine a microbial threshold with diagnostic utility in the routine diagnostic workup performed for suspected UTIs in the urology setting. The samples used in this study are representative of this routine diagnostic workup. Therefore, the samples included in this analysis contain a mix of cases positive and negative for microorganisms, as should be expected from samples sent for diagnostic testing. Cases with UTI-associated symptoms, including LUTS, and a presumptive UTI diagnosis that test negative for microorganisms may also represent alternative diagnoses such as overactive bladder or interstitial cystitis. While microorganism-negative samples are considered their own subgroup in our analyses, these samples were intentionally included in order to realistically represent the variety of samples that undergo microbial identification testing during diagnostic workup for presumed UTI.

References:

  1. Baunoch, D.; Luke, N.; Wang, D.; Vollstedt, A.; Zhao, X.; Ko, D.S.C.; Huang, S.; Cacdac, P.; Sirls, L.T. Concordance Between Antibiotic Resistance Genes and Susceptibility in Symptomatic Urinary Tract Infections. Infect Drug Resist 2021, 14, 3275–3286, doi:10.2147/idr.s323095.
  2. Wojno, K.J.; Baunoch, D.; Luke, N.; Opel, M.; Korman, H.; Kelly, C.; Jafri, S.M.A.; Keating, P.; Hazelton, D.; Hindu, S.; et al. Multiplex PCR Based Urinary Tract Infection (UTI) Analysis Compared to Traditional Urine Culture in Identifying Significant Pathogens in Symptomatic Patients. Urology 2020, 136, 119–126, doi:10.1016/j.urology.2019.10.018. 485
  3. Vollstedt; D, B.; KJ, W.; N, L.; K, C.; Belkoff, L.; Milbank, A.; N, S.; R, H.; N, G.; et al. Multisite Prospective Comparison of Multiplex Polymerase Chain Reaction Testing with Urine Culture for Diagnosis of Urinary Tract Infections in Symptomatic Patients. Journal of Surgical Urology 2020, doi:10.29011/ JSU-102.100002.
  4. Price, T.K.; Dune, T.; Hilt, E.E.; Thomas-White, K.J.; Kliethermes, S.; Brincat, C.; Brubaker, L.; Wolfe, A.J.; Mueller, E.R.; Schreckenberger, P.C. The Clinical Urine Culture: Enhanced Techniques Improve Detection of Clinically Relevant Microorganisms. J Clin Microbiol 2016, 54, 1216–1222, doi:10.1128/jcm.00044-16.
  5. Price, T.K.; Hilt, E.E.; Dune, T.J.; Mueller, E.R.; Wolfe, A.J.; Brubaker, L. Urine Trouble: Should We Think Differently about UTI? Int Urogynecol J 2018, 29, 205–210, doi:10.1007/s00192-017-3528-8.
  6. Harding, C.; Rantell, A.; Cardozo, L.; Jacobson, S.K.; Anding, R.; Kirschner-Hermanns, R.; Greenwell, T.; Swamy, S.; Malde, S.; Abrams, P. How Can We Improve Investigation, Prevention and Treatment for Recurrent Urinary Tract Infections – ICI-RS 2018. Neurourol. Urodyn. 2019, 38, S90–S97, doi:10.1002/nau.24021.
  7. Szlachta-McGinn, A.; Douglass, K.M.; Chung, U.Y.R.; Jackson, N.J.; Nickel, J.C.; Ackerman, A.L. Molecular Diagnostic Methods Versus Conventional Urine Culture for Diagnosis and Treatment of Urinary Tract Infection: A Systematic Review and Meta-Analysis. European Urology Open Sci 2022, 44, 113–124, doi:10.1016/j.euros.2022.08.009. 512
  8. Kline, K.A.; Lewis, A.L. Gram-Positive Uropathogens, Polymicrobial Urinary Tract Infection, and the Emerging Microbiota of the Urinary Tract. Microbiol Spectr 2016, 4, doi:10.1128/microbiolspec.uti-0012-2012.
  9. Lotte, R.; Lotte, L.; Ruimy, R. Actinotignum Schaalii (Formerly Actinobaculum Schaalii): A Newly Recognized Pathogen—Review of the Literature. Clin Microbiol Infec 2016, 22, 28–36, doi:10.1016/j.cmi.2015.10.038.
  10. Kaido, M.; Yasuda, M.; Komeda, H.; Okano, M.; Ito, Y.; Ohashi, H.; Ohta, H.; Akai, Y. Prediction of Presence of Fastidious Bacteria by the Fully Automated Urine Particle Analyzer UF-1000i in the Case of Ineffective Antimicrobial Therapy for Urinary Tract Infection. J Infect Chemother 2023, 29, 443–452, doi:10.1016/j.jiac.2023.01.009.
  11. Lewis DA, Brown R, Williams J, et al (2013) The human urinary microbiome; bacterial DNA in voided urine of asymptomatic adults. Frontiers in Cellular and Infection Microbiology 3:41. https://doi.org/10.3389/fcimb.2013.00041
  12. Gasiorek M, Hsieh MH, Forster CS (2019) Utility of DNA Next-Generation Sequencing and Expanded Quantitative Urine Culture in Diagnosis and Management of Chronic or Persistent Lower Urinary Tract Symptoms. J Clin Microbiol 58:. https://doi.org/10.1128/jcm.00204-19

Were there any significant differences between M-PCR and SUC in terms of their ability to detect microbes in urine samples?

Response: We appreciate your question. Our results showed that M-PCR has superior ability to detect microbes in urine samples than SUC. First, there were higher percentage of SUC specimens with negative results compared to M-PCR (Table 2). Please refer to our response to the previous question of yours for more detail in this aspect. Second, in samples where microbes were detected, M-PCR identified 883 microbes with densities at > 10,000 cells/mL, while SUC identified 496 samples with microorganisms at a threshold of > 10,000 CFU (Supplementary Table S2). Third, Supplemental Table S2 details the detections of gram-negative (E. coli and non-E. coli), gram-positive, fastidious bacteria, and yeasts, by M-PCR and SUC, respectively, demonstrating the powerful detection ability of M-PCR, compared to SUC for most microbes. For example, E. coli (gram-negative) was detected in 188 (32%) patients by M-PCR, vs. 160 (27%) patients by SUC, K. pneumoniae (gram-negative) was detected in 63 (11%) patients by M-PCR, vs. 59 (10%) patients by SUC, E. faecium (gram-positive) was detected in 10 (2%) patients by M-PCR, vs. 7 (1%) patients by SUC, A. urinae (fastidious) was detected in 118 (20%) patients by M-PCR, vs. 4 (1%) patients by SUC, and A. schaalii (fastidious) was detected in 120 (20%) patients by M-PCR, vs. 0 (0%) patients by SUC (Supplementary Table S2).

Our results are consistent with findings from multiple other studies, which provide additional evidence for the low sensitivity and significant number of false-negative results by SUC. (Refs 1-3)

  1. Heytens S, De Sutter A, Coorevits L, et al. Women with symptoms of a urinary tract infection but a negative urine culture: PCR-based quantification of Escherichia coli suggests infection in most cases. Clin Microbiol Infect. 2017;23(9):647-652. doi:10.1016/j.cmi.2017.04.004
  2. Naber KG, Schito G, Botto H, Palou J, Mazzei T. Surveillance study in Europe and Brazil on clinical aspects and Antimicrobial Resistance Epidemiology in Females with Cystitis (ARESC): implications for empiric therapy. Eur Urol. 2008;54(5):1164-1175. doi:10.1016/j.eururo.2008.05.010
  3. Barnes HC, Wolff B, Abdul-Rahim O, et al (2021) A Randomized Clinical Trial of Standard versus Expanded Cultures to Diagnose Urinary Tract Infections in Women. J Urology 206:1212–1221. https://doi.org/10.1097/ju.0000000000001949

Did the rise in biomarker levels correlate with increasing microbial density consistently for both M-PCR and SUC methodologies?

Response: We appreciate your question. Yes. Figure 1 and Table 2 (NGAL), Figure 2 and Table 3 (IL-8), and Figure 3 and Table 4 (IL-1β) in the manuscript showed that the medians, and in most cases, the means, of each of the three biomarker levels increased with the increase of microbial densities (<10,000 10,000-999,999, ≥ 100,000 cells/mL or CFUs/mL by M-PCR or SUC). Unlike SUC results, which were semiquantitative and thus not continuous, M-PCR microbial density results were quantitative. Therefore, we also conducted an in-depth correlation analysis further investigating this important question for M-PCR and recently submitted a separate manuscript detailing our results in this regard. Findings in this separate manuscript have demonstrated a significant positive correlation between microbial density detected by M-PCR and biomarker levels.

7.     What is the significance of the significantly higher number of SUC specimens with negative results compared to M-PCR? Are there any potential reasons for this discrepancy?

Response: Thank you for your questions. Yes, there was a higher percentage of SUC urine specimens (193/583, 33.1%) with negative results compared to M-PCR (117/583, 20.1%) (Table 2 of the manuscript). As described in the Discussion section of our manuscript (see lines 327-335), this observation is consistent with previous studies showing that M-PCR outperforms SUC in identifying and quantifying more causative uropathogens. (Refs 1-3)

Figure 5 of our manuscript also showed that SUC negative cases (n = 193) had biomarker consensus positive results in 46% of cases, which was much higher than M-PCR negative cases (n = 117) with 29% consensus positive. In another study (recently submitted) we conducted to compare biomarker levels in urine specimens in M-PCR-positive/SUC-negative and M-PCR-negative/SUC-negative patients, we found that median levels of all three biomarkers were significantly higher (p < 0.0001) in cases with M-PCR-positive/SUC-negative results (n = 80) than with M-PCR-negative/SUC-negative results (n = 107). Two or more biomarkers were positive in 76% of M-PCR-positive/SUC-negative specimens. These results suggest that SUC misses biomarker-positive UTIs detected by M-PCR.

The reason for this discrepancy is likely that the SUC method favors the growth of gram-negative microbes like E. coli but is less effective for non-E. coli microorganisms and fastidious microbes, which are increasingly recognized as common uropathogens.(Refs 5-16) Our Supplementary Table S2 details the detections of gram-negative (E. coli and non-E. coli), gram-positive, fastidious bacteria, and yeasts, by M-PCR and SUC, respectively, demonstrating again the powerful detection ability of M-PCR, compared to SUC for most microbes. Please refer to our response to Q5 of your comments for more details. Additionally, SUC is less likely to identify polymicrobial infections. (Refs 1-3)

References:

  1. Baunoch, D.; Luke, N.; Wang, D.; Vollstedt, A.; Zhao, X.; Ko, D.S.C.; Huang, S.; Cacdac, P.; Sirls, L.T. Concordance Between Antibiotic Resistance Genes and Susceptibility in Symptomatic Urinary Tract Infections. Infect Drug Resist 2021, 14, 3275–3286, doi:10.2147/idr.s323095.
  2. Wojno, K.J.; Baunoch, D.; Luke, N.; Opel, M.; Korman, H.; Kelly, C.; Jafri, S.M.A.; Keating, P.; Hazelton, D.; Hindu, S.; et al. Multiplex PCR Based Urinary Tract Infection (UTI) Analysis Compared to Traditional Urine Culture in Identifying Significant Pathogens in Symptomatic Patients. Urology 2020, 136, 119–126, doi:10.1016/j.urology.2019.10.018. 485
  3. Vollstedt; D, B.; KJ, W.; N, L.; K, C.; Belkoff, L.; Milbank, A.; N, S.; R, H.; N, G.; et al. Multisite Prospective Comparison of Multiplex Polymerase Chain Reaction Testing with Urine Culture for Diagnosis of Urinary Tract Infections in Symptomatic Patients. Journal of Surgical Urology 2020, doi:10.29011/ JSU-102.100002.
  4. Price, T.K.; Dune, T.; Hilt, E.E.; Thomas-White, K.J.; Kliethermes, S.; Brincat, C.; Brubaker, L.; Wolfe, A.J.; Mueller, E.R.; Schreckenberger, P.C. The Clinical Urine Culture: Enhanced Techniques Improve Detection of Clinically Relevant Microorganisms. J Clin Microbiol 2016, 54, 1216–1222, doi:10.1128/jcm.00044-16.
  5. Price, T.K.; Hilt, E.E.; Dune, T.J.; Mueller, E.R.; Wolfe, A.J.; Brubaker, L. Urine Trouble: Should We Think Differently about UTI? Int Urogynecol J 2018, 29, 205–210, doi:10.1007/s00192-017-3528-8.
  6. Harding, C.; Rantell, A.; Cardozo, L.; Jacobson, S.K.; Anding, R.; Kirschner-Hermanns, R.; Greenwell, T.; Swamy, S.; Malde, S.; Abrams, P. How Can We Improve Investigation, Prevention and Treatment for Recurrent Urinary Tract Infections – ICI-RS 2018. Neurourol. Urodyn. 2019, 38, S90–S97, doi:10.1002/nau.24021.
  7. Szlachta-McGinn, A.; Douglass, K.M.; Chung, U.Y.R.; Jackson, N.J.; Nickel, J.C.; Ackerman, A.L. Molecular Diagnostic Methods Versus Conventional Urine Culture for Diagnosis and Treatment of Urinary Tract Infection: A Systematic Review and Meta-Analysis. European Urology Open Sci 2022, 44, 113–124, doi:10.1016/j.euros.2022.08.009. 512
  8. Kline, K.A.; Lewis, A.L. Gram-Positive Uropathogens, Polymicrobial Urinary Tract Infection, and the Emerging Microbiota of the Urinary Tract. Microbiol Spectr 2016, 4, doi:10.1128/microbiolspec.uti-0012-2012.
  9. Lotte, R.; Lotte, L.; Ruimy, R. Actinotignum Schaalii (Formerly Actinobaculum Schaalii): A Newly Recognized Pathogen—Review of the Literature. Clin Microbiol Infec 2016, 22, 28–36, doi:10.1016/j.cmi.2015.10.038.
  10. Kaido, M.; Yasuda, M.; Komeda, H.; Okano, M.; Ito, Y.; Ohashi, H.; Ohta, H.; Akai, Y. Prediction of Presence of Fastidious Bacteria by the Fully Automated Urine Particle Analyzer UF-1000i in the Case of Ineffective Antimicrobial Therapy for Urinary Tract Infection. J Infect Chemother 2023, 29, 443–452, doi:10.1016/j.jiac.2023.01.009.
  11. Heytens S, De Sutter A, Coorevits L, et al. Women with symptoms of a urinary tract infection but a negative urine culture: PCR-based quantification of Escherichia coli suggests infection in most cases. Clin Microbiol Infect. 2017;23(9):647-652. doi:10.1016/j.cmi.2017.04.004
  12. Naber KG, Schito G, Botto H, Palou J, Mazzei T. Surveillance study in Europe and Brazil on clinical aspects and Antimicrobial Resistance Epidemiology in Females with Cystitis (ARESC): implications for empiric therapy. Eur Urol. 2008;54(5):1164-1175. doi:10.1016/j.eururo.2008.05.010
  13. Barnes HC, Wolff B, Abdul-Rahim O, et al (2021) A Randomized Clinical Trial of Standard versus Expanded Cultures to Diagnose Urinary Tract Infections in Women. J Urology 206:1212–1221. https://doi.org/10.1097/ju.0000000000001949
  14. Heytens S, De Sutter A, Coorevits L, et al. Women with symptoms of a urinary tract infection but a negative urine culture: PCR-based quantification of Escherichia coli suggests infection in most cases. Clin Microbiol Infect. 2017;23(9):647-652. doi:10.1016/j.cmi.2017.04.004
  15. Naber KG, Schito G, Botto H, Palou J, Mazzei T. Surveillance study in Europe and Brazil on clinical aspects and Antimicrobial Resistance Epidemiology in Females with Cystitis (ARESC): implications for empiric therapy. Eur Urol. 2008;54(5):1164-1175. doi:10.1016/j.eururo.2008.05.010
  16. Barnes HC, Wolff B, Abdul-Rahim O, et al (2021) A Randomized Clinical Trial of Standard versus Expanded Cultures to Diagnose Urinary Tract Infections in Women. J Urology 206:1212–1221. https://doi.org/10.1097/ju.0000000000001949

What are the implications of the significantly higher biomarker levels in the "no microorganisms detected" group when microbial detection was assessed by the SUC method compared to M-PCR? How do these findings impact the interpretation of the results?

Response: Thank you for your questions. We agree that the significantly higher biomarker levels in the SUC-negative compared to the M-PCR-negative group has important implications. As we described in the discussion section (see lines 335-352), there are several implications:

Firstly, the difference in biomarker levels underscores the potential variation in effectiveness between SUC and M-PCR as testing methods for detecting potential pathogens, suggesting that M-PCR may have higher sensitivity and specificity for detecting microbes that are causing a UTI, thereby enhancing diagnostic accuracy for symptomatic cases of UTI. Furthermore, SUC cases displaying low or no microbial densities alongside high biomarker levels suggest a potential failure in detecting the organisms causing the UTI leading to these elevated levels. These findings challenge the sensitivity of SUC for the identification of uropathogens, raising questions about false negatives in culture-based testing.

Consequently, the increase in biomarker levels between SUC-negative cases highlights the need to carefully consider the testing method employed when interpreting results and making treatment decisions, as SUC-negative cases may warrant closer monitoring.

Results in this manuscript are also consistent with findings from another recent publication by our group. We compared microbe detections in clinical samples by M-PCR, SUC, and expanded quantitative urine culture (EQUC) and showed that EQUC and M-PCR yielded very similar microbe detections. Of the 395 organisms detected by M-PCR, EQUC detected 89.1% (p = 0.10), whereas SUC only detected 27.3%. M-PCR identified 260 nonfastidious bacteria, EQUC detected 96.5% (p = 0.68), whereas SUC detected 41.5%. M-PCR identified 135 fastidious bacteria and EQUC 101 (74.8%, p = 0.01), whereas SUC failed to detect any (0%, p < 0.0001). These results indicate that most microbes identified by M-PCR, a significant portion of which missed by SUC, represented viable organisms, and validate M-PCR as a diagnostic tool for UTIs (Ref 1).

Reference:

  1. Festa R, Luke N , Mathur M, Parnell L, Wang D, Zhao X, Investigati JM, Chan M, Nguyen J, Cho T, Ngo A, Murphy M, Baunoch D, A test combining multiplex-PCR with pooled antibiotic susceptibility testing has high correlation with expanded urine culture for detection of live bacteria in urine samples of suspected UTI patients, Diagnostic Microbiology & Infectious Disease (2023), doi: https://doi.org/10.1016/j.diagmicrobio.2023.116015

How were the cutoffs for biomarker consensus established? Were they based on previous studies, clinical guidelines, or other criteria?

Response: Thank you for your question. As cited in the manuscript (see line 188), previously published thresholds for biomarker positivity were used as cutoffs for the analysis for NGAL (≥ 38 ng/mL), IL-8 (≥ 20.6 pg/mL), and IL-1β (≥ 12.4 pg/mL) in this study. (Refs 1,2)

References:

  1. Clinic, M. Neutrophil Gelatinase-Associated Lipocalin (NGAL) 2008.
  2. Yuan, Q.; Huang, R.; Tang, L.; Yuan, L.; Gao, L.; Liu, Y.; Cao, Y. Screening Biomarkers and Constructing a Predictive Model for Symptomatic Urinary Tract Infection and Asymptomatic Bacteriuria in Patients Undergoing Cutaneous Ureterostomy: A Metagenomic Next-Generation Sequencing Study. Dis Markers 2022, 2022, 7056517, doi:10.1155/2022/7056517.

What are the clinical implications of the study findings regarding the optimal microbial density threshold for diagnosing urinary tract infections? How might these findings impact current diagnostic practices?

Response: We appreciate your emphasis on these important implications and impacts. As described in the Discussion section (see lines 311-319), traditionally, a microbial density of ≥ 100,000 cells/mL has been deemed diagnostically significant; however, recent clinical reviews and guidelines have proposed lower thresholds that are still clinically relevant.(Refs1–6) In our study, we observed that lower microbial densities (≥ 10,000 cells/mL) detected by M-PCR and SUC in symptomatic subjects showed a statistically significant increase in infection-associated biomarker levels. Additionally, for these subjects with a suspected diagnosis of UTI, a microbial density of ≥ 10,000 cells/mL strongly correlated with biomarker consensus in both M-PCR and SUC techniques. These findings suggest that a microbial detection threshold of 10,000 cells/mL could be an indicative criterion for diagnosing a UTI. Using a threshold of >100,000 cells/mL as a criterion for initiating antimicrobial therapy carries significant clinical implications. One consequence is the potential for undertreatment of certain UTI patients, allowing microbes to proliferate, infiltrate host cells, develop biofilms, or acquire antibiotic resistance before treatment is administered.(Refs 7,8) Delaying treatment can lead to a progression of clinical severity, potentially necessitating higher antibiotic doses for extended durations or increasing the likelihood of complications, including recurrent infections and bacteremia.(Ref 9) Using a lower threshold of 10,000 cells/mL may improve diagnostic sensitivity and prevent or reduce these negative outcomes.

References:

  1. Rubin, R.H.; Shapiro, E.D.; Andriole, V.T.; Davis, R.J.; Stamm, W.E. Evaluation of New Anti-Infective Drugs for the Treatment of Urinary Tract Infection. Clin Infect Dis 1992, 15, S216–S227, doi:10.1093/clind/15.supplement_1.s216.
  2. Hovelius, B.; Mårdh, P.-A.; Bygren, P. Urinary Tract Infections Caused by Staphylococcus Saprophyticus: Recurrences and Complications. J Urology 1979, 122, 645–647, doi:10.1016/s0022-5347(17)56541-6.
  3. McNulty, C. PHE/NHS Diagnosis of Urinary Tract Infections Available online: https://assets.publishing.service.gov.uk/government/uploads/system/uploads/attachment_data/file/927195/UTI_diagnostic_flowchart_NICE-October_2020-FINAL.pdfov.uk) (accessed on 15 February 2023).
  4. Kouri, T.; Fogazzi, G.; Gant, V.; Hallander, H.; Hofmann, W.; Guder, W.G. European Urinalysis Guidelines. Scand J Clin Laboratory Investigation 2000, 60, 1–96, doi:10.1080/00365513.2000.12056993.
  5. Roberts, F.J. Quantitative Urine Culture in Patients with Urinary Tract Infection and Bacteremia. Am J Clin Pathol 1986, 85, 616–618, doi:10.1093/ajcp/85.5.616.
  6. Kunin Urinary Tract Infections: Detection, Prevention, and Management; Kunin, M., Ed.; Lea & Febiger: Philadelphia, 1997
  7. Mysorekar, I.; National, H.S. of the Mechanisms of Uropathogenic Escherichia Coli Persistence and Eradication from the Urinary Tract. 2006, doi:10.1073/pnas.0602136103.
  8. Holá, V.; Opazo-Capurro, A.; Scavone, P. Editorial: The Biofilm Lifestyle of Uropathogens. Front Cell Infect Mi 2021, 11, 763415, doi:10.3389/fcimb.2021.763415.
  9. Yang, X.; Chen, H.; Zheng, Y.; Qu, S.; Wang, H.; Yi, F. Disease Burden and Long-Term Trends of Urinary Tract Infections: A 503 Worldwide Report. Frontiers Public Heal 2022, 10, 888205, doi:10.3389/fpubh.2022.888205.7.

11. How do the limitations of the study, such as the inability to measure biomarker levels based on specific symptoms and the sample size, affect the generalizability and reliability of the results?

Response: We appreciate your question. In this study, we recruited subjects presenting at urology clinics in 39 U.S. states with clinical presentations and ICD-10-CM diagnostic codes consistent with UTI and aimed to compare their urine inflammation biomarker levels (IL-8, IL-1ß, and N-GAL) and microbial quantification results.  The determination of whether the clinical presentations and ICD-10-CM codes are consistent with UTI were made by the healthcare providers with urologic experience and expertise. These urologic specialists assigned the ICD-10-CM codes to the patients based on the patients' clinical presentations, which could include typical UTI symptoms defined by the FDA, such as urinary frequency, urinary urgency, dysuria, and suprapubic pain, Dipstick Urinalysis results, or other details. Therefore, in this study, we did not study the relationship between biomarker levels with specific symptoms. As mentioned in the Discussion section (see lines 374-377), the sample size also did not allow us to study relationships between biomarker levels and detection densities of specific microbial species or microbial combinations. Different microbial species may vary in their ability to elicit an immune response and ability to cause UTI, and the combined presence of multiple species can also influence the observed biomarker levels and pathogenesis toward UTI. Future work could explore the thresholds specific to individual or combinations of microbes and their pathogenic densities in detecting UTIs.

  1. Based on the conclusions of the study, what are the key takeaways for healthcare professionals and researchers in the field of urinary tract infections? What are the suggested areas for future research and clinical implementation?

Response: Thank you for your questions. Our study shows that symptomatic subjects with UTI exhibit a significant immune response at a microbial density threshold of > 10,000 cells/mL, regardless of the detection method used. The three major takeaways for healthcare professionals and researchers in the UTI field are:

First, a lower diagnostic microbial density threshold of 10,000 cells/mL is clinically appropriate for UTI diagnosis and management.

Second, our study emphasizes the need for closer monitoring when SUC yields a negative result, given its suboptimal microbial detection sensitivity and the resulting possibility of false negatives and their clinical implications.

Third, the diagnostic utility of the culture-independent M-PCR test, used in conjunction with clinical presentations, is supported by inflammation biomarker results.

As described in the discussion section (see lines 374-377), future studies could build upon these findings by assessing the biomarker levels across different species or combinations of species, allowing for a more targeted approach to diagnosing and managing UTIs. Additionally, exploring the association between biomarker levels and specific symptoms of UTI would provide valuable clinical information for personalized treatment strategies. These potential avenues of research can further enhance our understanding of UTIs and contribute to improved diagnostic accuracy and patient care.

Reviewer 3 Report

The research question including the mothodology were interesting.

1. The smaple size calculation should be addressed, how 583 pateiens from?

2. Urine wbc is the one that helpful and almost always investigated in UTI patients, this data might improved and provide more relavant data for clinicians.

3. The research question that synptomatic complicated UTI was very good. However, the author discussed about couldn't differentiate between the patients, symptoms. Fever might help for subgroup of Upper UTI and could have some more data to analyze.

4. The authors discussed of large smaple size is needed for further research. However, without sample size calculation addressed, could not expected how large is large enough.

Author Response

Comments and Suggestions for Authors

The research question including the methodology was interesting.

Note: During the revisions process, we discovered a transcription error in Tables 2-4 for the median and mean values for the SUC >100,000 CFUs/mL microbial density category. This error did not affect the figures, results, or conclusions text, and the tables have been corrected.

  1. The sample size calculation should be addressed, how 583 patients from?

Response: We appreciate your question regarding the sample size. We recruited subjects presenting at urology clinics in 39 U.S. states with clinical presentations and ICD-10-CM codes consistent with a presumed diagnosis of UTI and aimed to compare their urine inflammation biomarker levels (IL-8, IL-1ß, and N-GAL) and microbial quantification results.  Previous studies (ref 1,2, and 3) regarding the use of these biomarkers for the diagnosis of UTI, each included approximately 200 subjects. Between 01/17/2023 and 04/24/2023, our study was able to acquire 583 unique patient specimens, more than double the sample size of the other studies. Therefore, we stopped the enrollment on 04/25/23, and started the study analysis. This manuscript presents analysis results based on these 583 subjects.

References

  1. Gadalla AAH, Friberg IM, Kift-Morgan A, Zhang J, Eberl M, Topley N, et al. Identification of clinical and urine biomarkers for uncomplicated urinary tract infection using machine learning algorithms. Sci Rep-uk. 2019;9(1):19694. doi:10.1038/s41598-019-55523-x
  2. Tyagi P, Tyagi V, Qu X, Chuang YC, Kuo HC, Chancellor M. Elevated CXC chemokines in urine noninvasively discriminate OAB from UTI. Am J Physiol-renal. 2016;311(3):F548-F554. doi:10.1152/ajprenal.00213.2016
  3. Steigedal M, Marstad A, Haug M, Damås JK, Strong RK, Roberts PL, et al. Lipocalin 2 Imparts Selective Pressure on Bacterial Growth in the Bladder and Is Elevated in Women with Urinary Tract Infection. J Immunol. 2014;193(12):6081-6089. doi:10.4049/jimmunol.1401528

Urine WBC is the one that is helpful and almost always investigated in UTI patients, this data might improve and provide more relevant data for clinicians.

Response: Thank you for your suggestion. Indeed, leukocyte esterase (LE) dipstick analysis, a proxy for white blood cells, is often employed in clinics as part of the diagnostic workup for UTI. However, the specificity is generally low (specificity range is 9-59%, PPV is 86% and NPV is 72%.). The LE dipstick analysis is generally performed in a provider’s office before the clinician chooses to send a specimen for microbial identification. Our aim was to compare the performance of two microbial identification tests (SUC and M-PCR) which can be ordered at the clinician’s discretion, usually after an in-office LE dipstick test, to provide additional data on which to base treatment decisions.

References:

Horváth, J.; Wullt, B.; Naber, K.G.; Köves, B. Biomarkers in Urinary Tract Infections – Which Ones Are Suitable for Diagnostics and Follow-Up? Gms Infect Dis 8, Doc24, doi:10.3205/id000068.

Gadalla, A.A.H.; Friberg, I.M.; Kift-Morgan, A.; Zhang, J.; Eberl, M.; Topley, N.; Weeks, I.; Cuff, S.; Wootton, M.; Gal, M.; et al. Identification of Clinical and Urine Biomarkers for Uncomplicated Urinary Tract Infection Using Machine Learning Algorithms. Sci Rep-uk 2019, 9, 19694, doi:10.1038/s41598-019-55523-x.

Shaikh, N.; Martin, J.M.; Hoberman, A.; Skae, M.; Milkovich, L.; McElheny, C.; Hickey, R.W.; Gabriel, L.V.; Kearney, D.H.; Majd, M.; et al. Biomarkers That Differentiate False Positive Urinalyses from True Urinary Tract Infection. Pediatr Nephrol 2020, 35, 321–329, doi:10.1007/s00467-019-04403-7.

Schmiemann, G.; Kniehl, E.; Gebhardt, K.; Matejczyk, M.M.; Hummers-Pradier, E. The Diagnosis of Uri-nary Tract Infection. Deutsches Ärzteblatt Int 2010, 107, 361–367, doi:10.3238/arztebl.2010.0361.

  1. The research question that symptomatic complicated UTI was very good. However, the author discussed about couldn't differentiate between the patients, and symptoms. Fever might help for subgroups of Upper UTI and could have some more data to analyze.

Response: Thank you for your recommendation. A limitation of our use of biobanked urine specimens for this study is that we did not have access to specific symptom information (such as the presence of fever) for analysis. Further analyses in this area using data from upcoming clinical studies is a future direction we aim to pursue.

  1. The authors discussed of a large sample size is needed for further research. However, without sample size calculation addressed, could not be expected how large is large enough.

Response: We appreciate your feedback. Yes, we described in our discussion section that future studies could build upon the findings of the current study by assessing the biomarker levels across different species or combinations of species. The sample size needed for future studies of any specific microbial species will depend on the prevalence/positive rate of the microbial species in the target population as well as the goals of the specific future study.

Round 2

Reviewer 1 Report

The responses are convenient.

Reviewer 2 Report

The authors have addressed my queries and now the manuscript is acceptable for publication.